# Analysis of the Formation of Characteristic Aroma Compounds by Amino Acid Metabolic Pathways during Fermentation with *Saccharomyces cerevisiae*

**DOI:** 10.3390/molecules28073100

**Published:** 2023-03-30

**Authors:** Xingjun Lu, Chao Yang, Yingdi Yang, Bangzhu Peng

**Affiliations:** 1School of Public Health & Laboratory Medicine, Hunan University of Medicine, Huaihua 418000, China; 2College of Food Science and Technology, Huazhong Agricultural University, Wuhan 430070, China

**Keywords:** aroma compounds, correlation analysis, enzyme activity, fermentation, *Saccharomyces cerevisiae*

## Abstract

Amino acid metabolic pathways can have profound impacts on the activities of key enzymes in the biosynthesis of specific aroma compounds during yeast fermentation. Aroma compounds, pyruvic acid and glucose were monitored in relation to the key enzymes of leucine aminotransferase (LTR), phenylalanine aminotransferase (PAL), pyruvate kinase (PK) and acetyl-CoA in the amino acid metabolic pathways during the fermentation of simulated juice systems with added amino acids in order to explore the formation of characteristic aroma compounds. The addition of L-phenylalanine or L-leucine to the simulated juice systems significantly improved the activities of PK, PAL and LTR, and the content of acetyl-CoA, and significantly increased the concentrations of phenylethyl alcohol, octanoic acid, isoamyl acetate, phenylethyl acetate, ethyl hexanoate and ethyl caprylate during fermentation. Correlation analysis showed that there was a significant positive correlation between PAL, LTR, PK and acetyl-CoA and pyruvic acid formation. Path analysis revealed that the addition of amino acids affected the metabolism of pyruvate to alcohols, acids and esters to some extent.

## 1. Introduction

Amino acid metabolism is an important mechanism in the production of aroma compounds by *Saccharomyces cerevisiae* (*S. cerevisiae*) fermentation. Amino acids are common carbon sources, while α-amino acid is also the nitrogen source that yeasts mainly use [1]. Additionally, amino acids are one of the main precursor substances that can generate fermented flavor substances; the availability of amino acids can also affect many aspects of yeast metabolism during fermentation, during which yeasts convert amino acids to produce a large number of volatile and nonvolatile aroma compounds, including fatty acids, higher alcohols and esters, that are important for the organoleptic qualities of wines [2,3,4,5].

Several studies have revealed the effects of the addition of amino acids on the formation of volatile compounds [6,7,8]. Some branched-chain amino acids and aromatic amino acids are important precursors of aroma compounds. For example, phenylethanol is formed from L-phenylalanine by *Saccharomyces* yeasts and certain non-*Saccharomyces* yeasts, whereas isoamyl alcohol is derived from L-leucine [9,10,11].

Higher alcohols are produced from sugars and amino acids by yeast metabolism during alcoholic fermentation and can have an aromatic effect on wine products, especially phenethyl alcohol and isoamyl alcohol. Phenethyl alcohol, with its unmistakable odor of roses, is known to have a positive influence on wine aroma [12]. The formation of phenethyl alcohol is achieved by two pathways in the fermentation process: the shikimate pathway and the Ehrlich pathway. The shikimate pathway is a metabolic pathway for the de novo synthesis of phenethyl alcohol by *S. cerevisiae*: phosphoenolpyruvate (derived from the glycolytic pathway) and 4-phospho-erythrose (derived from the pentose phosphate pathway) form a shikimic acid by enzymatic reactions; then, the shikimic acid forms a phenylpyruvate and finally produces phenethyl alcohol under the action of phenylpyruvate decarboxylase and alcohol dehydrogenase [9]. In the Ehrlich pathway, the commonly accepted route from L-phenylalanine to phenethyl alcohol is by transamination of the amino acid to phenylpyruvate, decarboxylation to phenylacetaldehyde and reduction to phenethyl alcohol [9]. Moreover, *S. cerevisiae* inherently produces isoamyl alcohol through the breakdown of L-leucine via the Ehrlich pathway [13]. These alcohols can be converted into esters by *Saccharomyces* yeasts due to the action of alcohol acetyltransferases in the presence of acetyl Co-A. Acetate esters, such as isoamyl acetate and 2-phenylethyl acetate, are recognized as important flavor compounds in wine, imparting characteristic flavors [14].

In addition, the key enzymes in amino acid metabolic pathways have some vital effects on the formation of specific aroma compounds during *S. cerevisiae* fermentation [15,16]. However, the comprehensive roles of these key enzymes and their contribution to the production of aroma compounds have not been explored thoroughly, and it is of interest to understand the formation of aroma compounds by amino acid metabolism during fermentation. In this study, L-phenylalanine and L-leucine were added into a simulated juice system, the key enzyme activities and aroma compound concentrations were determined during fermentation with *S. cerevisiae*, and then correlation analysis was performed on the key enzymes of leucine aminotransferase (LTR), phenylalanine aminotransferase (PAL), pyruvate kinase (PK), acetyl-CoA and the characteristic aroma compound content. Finally, amino acid metabolic pathways were analyzed to explore the formation of characteristic aromas during fermentation.

## 2. Results and Discussion

### 2.1. Changes in the Content of Reducing Sugar and Pyruvic Acid

In this study, the changes in glucose and pyruvic acid during fermentation by *S. cerevisiae* CICC 31,084 in simulated juice systems with the addition of different amino acids were monitored, and the results are shown in Figure 1. The result showed that the rate of glucose metabolism and the rate of pyruvic acid synthesis were low for the first 12 h, then increased rapidly after 24 h in the amino acid-added simulated juice systems. The concentration of pyruvic acid reached a maximum at 72 h in the leucine-added simulated juice system and 48 h in the phenylalanine-added simulated juice system. By comparing the changes in the pyruvic acid content of the simulated juice systems with the result reported by Yang et al. [17], the maximum pyruvic acid content increased significantly with the addition of amino acids, reaching 0.0996 g/L at 72 h with L-leucine added and 0.0867 g/L at 48 h with L-phenylalanine added. After the maximum was reached, pyruvic acid production decreased significantly, which could be caused by *S. cerevisiae* using the keto acid produced by glucose metabolism to synthesize the required amino acids for itself [18,19]. The results indicated that the addition of an amino acid has a significant effect on pyruvic acid metabolism during fermentation, which improved pyruvic acid synthesis.

### 2.2. Changes in the Content of Characteristic Aroma Components

Changes in the content of characteristic aroma compounds of the simulated juice systems with the addition of different amino acids during fermentation are shown in Figure 2. The contents of isoamyl alcohol and isoamyl acetate were much higher than other aroma substances in the leucine-added simulated juice system, indicating that the addition of leucine promoted the formation of isoamyl alcohol and isoamyl acetate. Similarly, the contents of phenylethyl alcohol and phenylethyl acetate were much higher than other aroma substances in the phenylalanine-added simulated juice system during fermentation. This observation indicated that the addition of phenylalanine promoted the formation of phenylethyl alcohol and phenylethyl acetate, and this result was consistent with that of Trinh et al. [4].

In terms of the production of phenylethyl alcohol (Figure 2a), its content was the largest in the phenylalanine-added simulated juice system (maximum value of 34.12 mg/L reached at 72 h), followed by the leucine-added simulated juice system (maximum value of 0.89 mg/L reached at 120 h), and finally the simulated juice system without amino acid addition (maximum value of 0.14 mg/L reached at 120 h). As shown in Figure 2b,c, 1-pentanol was not detected in the leucine-added simulated juice system, but the content of isoamyl alcohol was extremely high (maximum value of 24.15 mg/L reached at 120 h). This result may occur because 1-pentanol and isoamyl alcohol are isomers of pentanol. In the phenylalanine-added and no amino acid-added simulated juice systems, isoamyl alcohol was not detected, while 1-pentanol was detected. The content of 1-pentanol was significantly higher in the phenylalanine-added simulated juice system (maximum value of 0.78 mg/L reached at 72 h) than in the no amino acid simulated juice system (maximum value of 0.18 mg/L reached at 72 h). The results show that the addition of amino acids significantly increased the formation of alcohols, and different amino acids promoted the formation of different alcohols; phenylalanine, a precursor of phenylethyl alcohol, significantly increased the production of phenylethyl alcohol, and leucine, as a precursor of isoamyl alcohol, obviously promoted the content of isoamyl alcohol. It is worth noting that the presence of leucine also promoted the formation of phenylethyl alcohol. This finding was consistent with the results of Lilly et al. [3].

The contents of octanoic acid in the different simulated juices systems are shown in Figure 2d. The maxima of octanoic acid were 1.76 mg/L at 72 h, 1.99 mg/L at 120 h and 0.28 mg/L at 96 h, in phenylalanine-added, leucine-added and no amino acid simulated juice systems during fermentation, respectively. Obviously, the contents of octanoic acid in the simulated juice systems with the addition of amino acids were much higher than in the simulated juice system with no amino acids. The results indicate that the addition of amino acids significantly promoted the formation of octanoic acid, and various amino acid species led to differences in the time at which octanoic acid reached the maximum value. The addition of leucine has an inhibitory effect on the formation of octanoic acid, which may lead to a later time for reaching its maximum [20]. As the content of assimilable nitrogen increased, the production of medium-chain fatty acids, such as octanoic acid, increased. Trinh et al. [4] reported that the addition of phenylalanine (0.05%, *w*/*v*) made no significant difference to the synthesis of octanoic acid, which was different from our results. This may have occurred because the 2.0 g/L (0.20%, *w*/*v*) amino acid addition in this study largely promoted octanoic acid synthesis during fermentation by *S. cerevisiae*.

Changes in the characteristic esters (phenylethyl acetate, isoamyl acetate, ethyl hexanoate and ethyl caprylate) of the simulated juice systems with/without the addition of amino acids during fermentation are shown in Figure 2e–g, and h, respectively. In the phenylalanine-added, leucine-added and no amino acid simulated juice systems, the maximum values of phenylethyl acetate were 18.48 mg/L, 0.43 mg/L and 0.04 mg/L, respectively; the maximum values of isoamyl acetate were 0.27 mg/L, 12.18 mg/L and 0.06 mg/L, respectively; the maximum values of ethyl hexanoate were 0.65 mg/L, 2.98 mg/L and 0.29 mg/L, respectively; and the maximum values of ethyl caprylate were 1.10 mg/L, 7.46 mg/L and 0.42 mg/L, respectively. Obviously, the addition of amino acids promoted the formation of these esters. Higher content of assimilable nitrogen in fermentation broth led to higher production of medium-chain fatty acid ethyl esters and acetates. In the phenylalanine-added simulated juice system, the contents of the characteristic esters in descending order were phenylethyl acetate, ethyl caprylate, ethyl hexanoate and isoamyl acetate; in the leucine-added simulated juice system, the contents of characteristic esters from high to low were isoamyl acetate, ethyl caprylate, ethyl hexanoate and phenylethyl acetate. It could be seen from the comparison that different amino acids promoted the formation of different esters; phenylalanine significantly promoted the formation of phenylethyl acetate, while inhibiting the formation of isoamyl acetate; with leucine, the opposite was the case. It was speculated that the synthesis of phenylethyl acetate and isoamyl acetate requires the participation of an acetyl group and corresponding alcohols, and the acetyl group is a precursor shared by both. There is a competitive relationship between the synthesis of phenylethyl acetate and isoamyl acetate when the acetyl group content in the system is limited. When phenylalanine content is higher in the simulated juice system, the content of phenylethyl alcohol is higher (Figure 2a), which is favorable for the formation of phenylethyl acetate (Figure 2e). When the leucine content is higher in the simulated juice system, a large amount of isoamyl alcohol will be produced (Figure 2b), which facilitates the formation of isoamyl acetate (Figure 2f). It could also be seen that the addition of phenylalanine in the simulated juice system can effectively promote the synthesis of phenylethyl hexanoate (Figure 2i) and phenyl caprylate (Figure 2j), which were not detected in the simulated systems with the addition of leucine and without the addition of amino acids. Therefore, there were significant differences between aroma substances produced by *S. cerevisiae* in simulated juice systems with the addition of different amino acids.

### 2.3. Changes in the Activities of Different Key Enzymes

Changes in the activities of leucine aminotransferase (LTR), phenylalanine aminotransferase (PAL) and pyruvate kinase (PK) and the concentration of acetyl-CoA in the simulated juice system with the addition of different amino acids during fermentation are shown in Figure 3. The LTR uses leucine and α-ketoglutarate as substrates to produce α-ketoisocaproic acid and glutamic acid through a transamination reaction [21]. Phenylalanine and α-ketoglutarate react in the presence of PAL to form α-phenylpyruvate and glutamate [9]. As seen in Figure 3a and b, the activities of LTR and PAL gradually increased at the start of fermentation and reached a maximum at approximately 84 h and 60 h, respectively. PK and acetyl CoA are important enzymes in the process of glycolysis [17]. As shown in Figure 3c, the PK activity in both simulated juice systems with the addition of amino acids showed the same tendency of increasing at the beginning and then declining later, which was similar to the simulated system without amino acids [17]. However, the rate of increase of PK activity was higher in the phenylalanine-added simulated juice system than in the leucine-added system. The PK activities reached a maximum of 88.56 U/g prot. at 72 h in the phenylalanine-added simulated juice system and 74.65 U/g at 108 h in the leucine-added simulated juice system. As shown in Figure 3d, the accumulation of acetyl-CoA was different in the simulated juice system with the addition of different amino acids. The content of acetyl-CoA in the phenylalanine-added simulated juice system increased rapidly and reached a maximum of 6.95 nmol/mg prot. at 48 h. The content of acetyl-CoA in the leucine-added simulated juice system increased slowly during the first 72 h, and then increased with fluctuations, reaching a maximum of 8.44 nmol/mg at 108 h.

### 2.4. Correlation Analysis of the Key Enzymes and Aroma Components

A correlation analysis was performed on the key enzyme activities and aroma components to explore the relationship between the key enzymes and aroma components produced by the amino acid metabolic pathways during the fermentation of *S. cerevisiae*, and the results of the leucine-added and phenylalanine-added simulated juice systems are shown in Table 1 and Table 2, respectively.

In the leucine-added simulated juice system (Table 1), LTR had a significant positive correlation with pyruvic acid, isoamyl acetate and ethyl hexanoate (*p* < 0.05). PK had a significantly negative correlation with glucose and leucine but a positive correlation with octanoic acid, alcohols (isoamyl alcohol, phenylethyl alcohol) and esters (isoamyl acetate, phenylethyl acetate, ethyl hexanoate, ethyl caprylate, ethyl caprate) (*p* < 0.05). Acetyl-CoA had a significantly positive correlation with leucine, octanoic acid, isoamyl alcohol, phenylethyl alcohol and ethyl caprate (*p* < 0.05). Both acetyl-CoA and acetic acid can provide acetyl groups for phenylethyl acetate and isoamyl acetate, but the correlations between acetyl-CoA and these two acetates were not high. It is speculated that the pathway of esters synthesis by acetyl-CoA is not the main biosynthesis pathway for phenylethyl acetate and isoamyl acetate [22].

In the phenylalanine-added simulated juice system (Table 2), PAL had a significantly positive correlation with pyruvic acid, hexanoic acid, phenylethyl alcohol and esters (ethyl hexanoate, phenylethyl hexanoate, phenylethyl caprylate, phenylethyl acetate, isoamyl acetate) (*p* < 0.05). PK had a significantly negative correlation with glucose and phenylalanine but a positive correlation with acids (octanoic acid, hexanoic acid), alcohols (1-pentanol, phenylethyl alcohol) and esters (ethyl caprylate, ethyl hexanoate, phenylethyl hexanoate, phenylethyl caprylate, phenylethyl acetate, isoamyl acetate) (*p* < 0.05). There were also significantly positive correlations between acetyl-CoA and pyruvic acid (*p* < 0.01).

Pyruvic acid had a significantly positive correlation with LTR and PAL, as shown in Table 1 and Table 2, respectively (*p* < 0.01), indicating that the change trends in these enzyme activities were consistent with that of pyruvic acid content. However, there was no significant correlation between pyruvic acid and PK (*p* > 0.05), which is different from the result of a significant correlation between PK and pyruvic acid in the simulated juice system without the addition of an amino acid reported by Yang et al. [17]. This finding shows that after the addition of amino acids, the production of pyruvic acid was affected not only by glucose metabolism but also by amino acid metabolism. The addition of amino acids as substrates into the simulated juice systems made a more complicated metabolic pathway and indistinctive correlations between aroma compounds and a single enzyme.

### 2.5. Path Analysis

Path analysis, which is used to study the linear relationship between multiple variables, has been widely used in many fields. Path analysis can show the relative importance of dependent variables by correlating the independent variables and dependent variables. Path analyses of the key compounds produced by *S. cerevisiae* in the leucine-added and phenylalanine-added simulated juice systems are shown in Figure 4. As can be seen in Figure 4, the higher path coefficient between glucose and pyruvic acid in both simulated juice systems indicated that the effect of glucose on pyruvic acid was crucial; conversely, the path coefficients between leucine, phenylalanine and pyruvic acid were low. This result may occur because the transformation process from amino acid to pyruvic acid was a long reaction path and involved reversible reactions. The path coefficients between isoamyl alcohol and isoamyl acetate, phenethyl alcohol and phenylethyl acetate, phenethyl alcohol and phenylethyl caprylate, and phenethyl alcohol and phenylethyl hexanoate were higher, indicating that alcohols were important precursors for esters and had strong influences on the formation of esters. These results suggest that the addition of amino acids made the metabolic pathways more complicated and affected the metabolic pathway of pyruvate to alcohols, acids and esters to some extent.

## 3. Materials and Methods

### 3.1. Reagents

L-Leucine and L-phenylalanine were purchased from Ding Guo Prosperous Co., Ltd. (Beijing, China). Peptone and yeast extract were supplied by Aobo Xing Biological Co., Ltd. (Beijing, China). Nitrogen base without amino acids (YNB) and O-phthalaldehyde (OPA) were obtained from Becton and Dickinson Difco Co., Ltd. (Franklin Lakes, NJ, USA). 3-Octanol was provided by TCI-Tixi Ai Chemical Industry Development Co., Ltd. (Shanghai, China). Other reagents were purchased from Sinopharm Chemical Reagent Co., Ltd. (Beijing, China). The phenylalanine aminotransferase (PAL) assay kit, leucine aminotransferase (LTR) activity assay kit and acetyl-CoA content assay kit were obtained from Sino Best Biological Technology Co., Ltd. (Shanghai, China). The pyruvate kinase (PK) assay kit, coomassie brilliant blue protein quantitative test kit and bicinchoninic acid (BCA) protein quantitative test kit were acquired from the Nanjing Institute of Bioengineering (Nanjing, China).

### 3.2. Preparation of Analytical Samples

The analytical samples were prepared as follows: 50 mL activated yeast culture solution with a concentration of 1.5 × 10^6^ cfu/mL (*S. cerevisiae* CICC 31084, purchased from the China Center of Industrial Culture Collection) was centrifuged at 3000 r/min for 10 min to collect the yeast cells. These cells were washed with sterile phosphate buffer saline (PBS) 2–3 times. Then, the as-prepared cells were dissolved by 750 mL sterilized simulated juice system (2.0 g/L L-leucine or L-phenylalanine, 100.0 g/L glucose, and 20.0 g/L YNB, and the pH was adjusted to 3.4 with HCl) with an inoculation concentration of 1.0 × 10^5^ cfu/mL. The mixture system was prepared in triplicate and fermented in an incubator at 28 °C. The samples were collected every 12 h and stored at −20 °C for the following analysis.

### 3.3. Analytical Determinations

The content of amino acids was determined according to the methods of Hu et al. [21] with some modifications. Samples were diluted 2–4 times. Then, 100 μL of the dilution was shaken well with 300 μL of O-pathaldialdehyde (OPA) reagent and 600 μL borate buffer in a 1.5 mL centrifuge tube at room temperature. All the derived mixtures were filtrated in sampler vials through a 0.45 μm membrane millipore filter, incubated for exactly 15 min in the dark and promptly sampled for high-performance liquid chromatographic determination by a Waters 1525 liquid chromatograph, consisting of a 2475 fluorescence detector. Mobile phase A was 25 mmol/L pH 6.0 sodium acetate buffer/tetrahydrofuran (95/5, *v*/*v*) and mobile phase B was absolute methanol. Fluorescence detection was performed with excitation and emission wavelengths of 340 nm and 450 nm, respectively. A Waters Symmetry^®^ C18 column (250 mm × 4.6 mm, 5 μm) was used for amino acid separation. The column temperature was set to room temperature, the flow rate of the mobile phase was 0.8 mL/min, and the injection volume was 5 μL.

The content of reducing sugar was determined by the 3, 5-dinitrosalicylic acid (DNS) method of a previous protocol and with some modifications [23]. A 1.0 mL fermentation broth was diluted 10 times and 1.0 mL dilution was mixed well with 1.5 mL of DNS solution. The mixture was placed in a boiling water bath for 5 min and diluted with distilled water to 15 mL after cooling to room temperature. Then its absorbance was detected at 540 nm.

Pyruvic acid content was determined by the 2,4-dinitrophenylhydrazine (DNPH) method, which referred to Tang [20] and with slight modifications. Pyruvic acid was used as the reference standard. The 1.0 mL fermentation broth, incubated for different times, was added to 2.0 mL of 8% (*w*/*v*) trichloroacetic acid (TCA) and mixed with 1 mL of 0.1% (*w*/*v*) DNPH reagent, then mixed well with 4.0 mL of 2 mol/L NaOH. After 10 min chromogenic reaction, the absorbance of the mixture was measured at 520 nm.

The aroma compounds of simulated juices with different fermented times were analyzed by HS-SPME/GC-MS [17]. A 4.8 mL sample of each fermentation broth fermented for different lengths of time was placed into a 15 mL SPME glass vial together with 200 μL 3-octyl alcohol, and 2.0 g of sodium chloride. The 3-octyl alcohol was the internal standard for the semiquantification of volatile compounds and its final concentration was 32.72 μg/L in the sample. After the sample was equilibrated in a water bath at 45 °C for 10 min, a fiber extraction head (50/30 μm DVB/CAR/PDMS, Supelco Co., Bellefonte, PA, USA) was inserted into the sample bottle to adsorb for 35 min. Then, the extraction head was injected into a gas chromatography-mass spectroscopy (GC-MS) system (Agilent 6890N/5975B, Agilent Technologies, Santa Clara County, CA, USA) and desorbed at 250 °C for 5 min. The fraction components of each sample were analyzed by HP-5MS elastic quartz capillary column (30 m × 0.25 mm × 0.25 μm) with helium at 1.0 mL/min in unsplit-flow mode. The temperature program of the column was 40 °C for 5 min, increase to 230 °C at a rate of 3 °C/min, then an 8 °C/min ramp to 230 °C and holding for 5 min. The mass spectrometer conditions were set as follows: ion source temperature, 230 °C; ion energy for electron impact (EI), 70 eV; quadrupole temperature, 150 °C; scan range, 50–550 AMU. The qualitative analysis of the identified volatile components was referred to the spectra in the NIST (National Institute of Standards and Technology, Gaithersburg, Maryland) library. The peak area ratio of internal standard substance and aroma components was used to calculate the concentration of aroma, and the formula was as follows:*C*_1_ = *P*_1_ × *C*_0_/*P*_0_
where *C*_1_ and *C*_0_ are the concentration of aroma components and 3-octanol (μg/L), respectively, and *P*_1_ and *P*_0_ are the peak area of the aroma component and 3-octanol, respectively.

The activity of phenylalanine aminotransferase (PAL), leucine aminotransferase (LTR), pyruvate kinase (PK), and acetyl-CoA content were determined according to the instructions of the Nanjing Jiancheng reagent kit [17].

### 3.4. Data Analysis

In this study, all the experiments were conducted in triplicate, and the data were reported as mean ± standard deviation. SPSS Statistics 22.0 software (IBM) was used to determine the difference, the data correlation and the path analysis. The differences at *p* < 0.05 were considered significant, and *p* < 0.01 were extremely significant. Path analysis was used to investigate the importance of the influence of different precursors on the same product. The number on the arrow shows the path coefficient. When the absolute value of the path coefficient is larger, the direct influence of the independent variable on the dependent variable is larger.

## 4. Conclusions

The addition of amino acids to simulated juice systems significantly improved the rates of glucose consumption and pyruvic acid production during the early stage of fermentation by *S. cerevisiae*. The activities of PK and aminotransferase and the content of acetyl-CoA increased significantly with the rapid consumption of glucose and amino acids during the middle stage of fermentation. The concentrations of phenylethyl alcohol, octanoic acid, isoamyl acetate, phenylethyl acetate, ethyl hexanoate and ethyl caprylate during fermentation increased significantly during fermentation in the amino acid-added simulated juice systems. The contents of isoamyl alcohol and isoamyl acetate increased much more substantially than the other aroma compounds due to the addition of L-leucine, and the contents of phenylethyl alcohol and phenylethyl esters increased much more substantially due to the addition of L-phenylalanine.

Correlation analysis for the leucine-added simulated juice system showed that LTR had a significantly positive correlation with the formation of pyruvic acid, isoamyl acetate and ethyl hexanoate. PK had a significant correlation with glucose, leucine, octanoic acid, alcohols and esters. Acetyl-CoA also had a significantly positive correlation with leucine, octanoic acid, isoamyl alcohol, phenylethyl alcohol and ethyl caprate. Moreover, correlation analysis for the phenylalanine-added simulated juice system showed that PAL had a significantly positive correlation with pyruvic acid, hexanoic acid, phenylethyl alcohol and most esters. PK had a significant correlation with glucose, phenylalanine, acids, alcohols and esters.

Path analysis results showed that the addition of amino acids significantly increased the path coefficient between pyruvic acid and isoamyl acetate but sharply reduced the path coefficient between pyruvic acid and phenylethyl acetate, and thus affected the metabolic pathway of pyruvate to alcohols, acids and esters to some extent. In general, the addition of amino acids affects the aroma composition and the activity of key enzymes in the fermentation process of *S. cerevisiae*, which provides a theoretical basis for improving the aroma composition of wine.

## Figures and Tables

**Figure 1 molecules-28-03100-f001:**
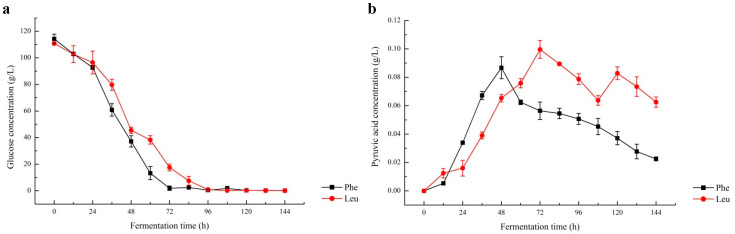
Changes in glucose (**a**) and pyruvic acid (**b**) of the simulated juice system with the addition of L-Leucine (
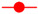
) and L-phenylalanine (
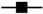
), respectively, during fermentation.

**Figure 2 molecules-28-03100-f002:**
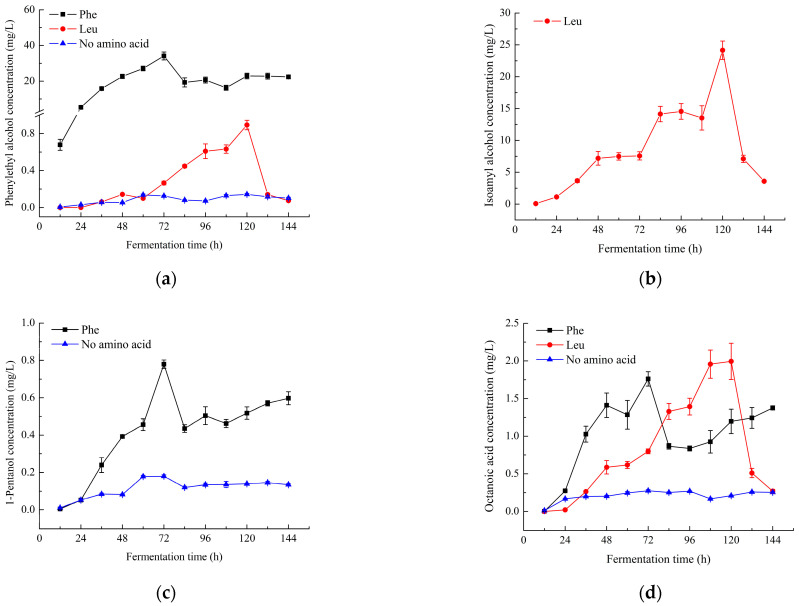
The changes in the content of the same characteristic aroma compound during fermentation in the simulated juice systems with/without the addition of amino acids: (**a**) phenylethyl alcohol; (**b**) isoamyl alcohol; (**c**) 1-pentanol; (**d**) octanoic acid; (**e**) phenylethyl acetate; (**f**) isoamyl acetate; (**g**) ethyl hexanoate; (**h**) ethyl caprylate; (**i**) phenylethyl hexanoate; (**j**) phenyl caprylate.

**Figure 3 molecules-28-03100-f003:**
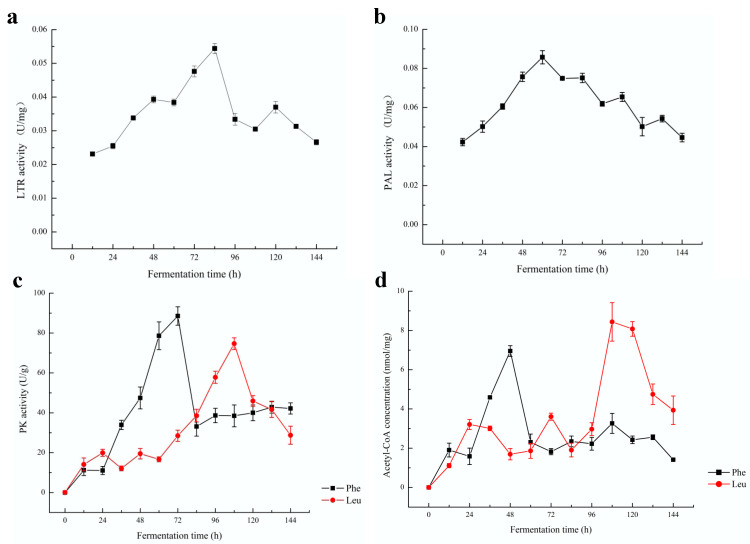
Changes in the activities of key enzymes in the simulated juice systems with the addition of different amino acids during fermentation. (**a**) LTR in the leucine-added simulated juice system; (**b**) PAL in the phenylalanine-added simulated juice system; (**c**) PK; (**d**) acetyl-CoA.

**Figure 4 molecules-28-03100-f004:**
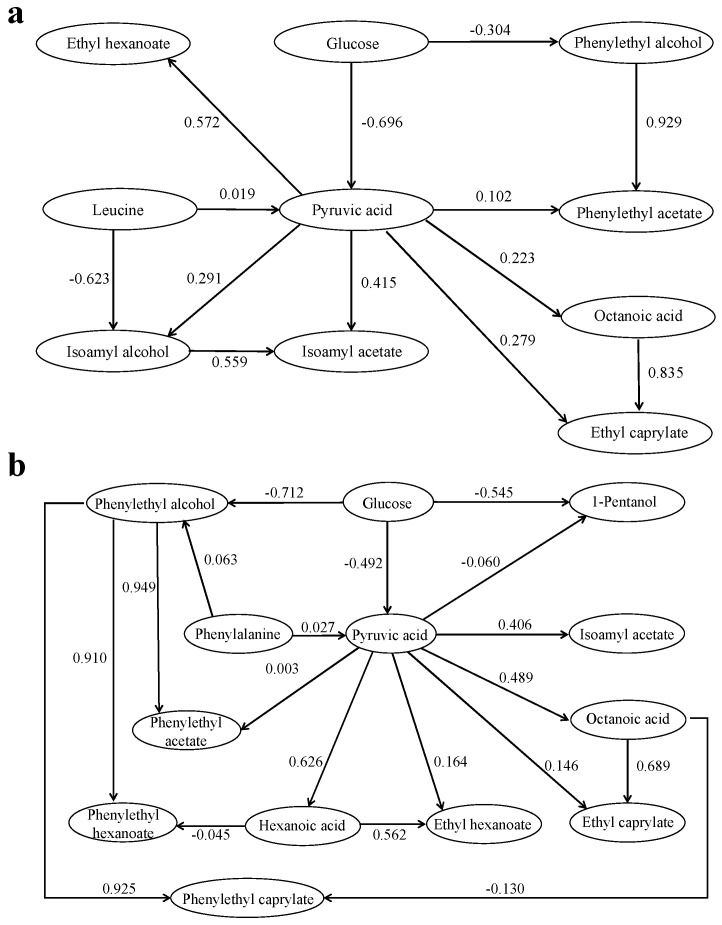
Path analysis on key compounds in simulated juice systems with the addition of amino acids during fermentation. (**a**) Leu-added; (**b**) Phe-added.

**Table 1 molecules-28-03100-t001:** Correlation analysis of the key enzymes and aroma components in the leucine-added simulated juice system.

	Isoamyl Alcohol	Isoamyl Acetate	Ethyl Hexanoate	Phenylethyl Alcohol	Phenylethyl Acetate	Octanoic Acid	Ethyl Caprylate	Ethyl Caprate	Pyruvic Acid	Glucose	Leucine
LTR	0.452	0.752 **	0.669 *	0.321	0.511	0.4	0.494	0.152	0.758 **	−0.395	−0.376
PK	0.687 *	0.623 *	0.669 *	0.798 **	0.759 **	0.834 **	0.783 **	0.599 *	0.444	−0.732 **	−0.765 **
Acetyl-CoA	0.610 *	0.238	0.321	0.685 *	0.564	0.688 *	0.613 *	0.749 **	0.254	−0.53	−0.592 *

Note: ** indicates a significant correlation at the 0.01 level; * indicates a significant correlation at the 0.05 level.

**Table 2 molecules-28-03100-t002:** Correlation analysis of the key enzymes and aroma components in the phenylalanine-added simulated juice system.

	1-Pentanol	Isoamyl Acetate	Hexanoic Acid	Ethyl Hexanoate	Phenylethyl Hexanoate	Phenylethyl Alcohol	Phenylethyl Acetate	Octanoic Acid	Ethyl Caprylate	Phenylethyl Caprylate	Pyruvic Acid	Glucose	Phenylalanine
PAL	0.377	0.791 **	0.610 *	0.897 **	0.773 **	0.599 *	0.718 **	0.503	0.515	0.690 *	0.817 **	−0.384	−0.389
PK	0.777 **	0.915 **	0.730 **	0.864 **	0.851 **	0.908 **	0.957 **	0.845 **	0.704 *	0.882 **	0.488	−0.611 *	−0.613 *
Acetyl-CoA	−0.09	0.163	0.355	0.278	0.321	0.093	0.153	0.252	−0.004	−0.075	0.722 **	0.082	0.065

Note: ** indicates a significant correlation at the 0.01 level; * indicates a significant correlation at the 0.05 level.

## Data Availability

Data is available upon reasonable request from the corresponding authors.

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
