# Peer review of "Analysis of the Formation of Characteristic Aroma Compounds by Amino Acid Metabolic Pathways during Fermentation with Saccharomyces cerevisiae"

_molecules, 2023, doi:10.3390/molecules28073100_

Round 1

Reviewer 1 Report

Peer review for Lu et al.“Analysis on formation of characteristic aroma compound by amino acid metabolic pathways during fermentation with Saccharomyces cerevisiae

Summary:

This article reviews presents analysis of key aroma compounds during fermentation of S. cerevisiae. Specifically, the authors examine the effect of amino acid addition (leucine and phenylalanine) on the concentration of aroma compounds and the activity of associated biosynthetic enzymes. Overall, I found the motivation for this work somewhat unclear, and the findings lack interpretation. A few pieces of data are also missing. This article must undergo the following modifications before it is fit for publication in Molecules:

Major comments:

1.      The last paragraph of the introduction lacks detail regarding the role of LTR, PL, and PK in the pathway for production of aroma compounds. A figure showing the structures of these aroma compounds and pathways for their biosynthesis from amino acids would be extremely useful to frame the study and provide a reference for readers as the findings of the paper are presented.

2.      The results should be discussed in the context of aroma compound biosynthesis pathways. Mechanistic explanations for differences in aroma compound quantity and/or enzyme activity should be explained using knowledge of the metabolic pathways.

3.      YNB with glucose and amino acids differs substantially from the composition of grape juice for wine. The term “simulated juice” should either be replaced, or the authors should demonstrate that the composition of this medium is indeed similar to grape juice.

4.      Figure 1 is lacking data and must be revised. Changes in glucose and pyruvic acid were measured with the addition of Leu and Phe during fermentation, but there is no negative control that measures these compounds without amino acid addition. Each of these plots should contain a third set of datapoints for glucose and pyruvic acid quantitation during fermentation without Leu or Phe added to the broth (as in Figure 2a, for example).

5.      Similar to the previous point, Figure 3 is also difficult to interpret without measurements of enzyme activity in all three conditions (Phe, Leu, and No amino acid). These data should be added to these plots for completeness.

6.      This study would also benefit from more discussion about the applications and benefits of understanding characteristic aroma compound formation in S. cerevisiae. For example, do the findings from this paper pave the way for metabolic engineering efforts in S. cerevisiae to enhance aroma compound production? How would this be leveraged in wine making?

Minor comments:

1.      Lines 115: “The 2,4-dinitrophenylhydrazine (DNPH) method was determined the content…” Please remove the word “was” – it is incorrect grammar.

2.      Line 116: “The pyruvic acid…” Please remove the word “The” – it is incorrect grammar.

3.      Line 177: “The 1.0 mL fermentation broth of different times…” Is also poorly worded. Please revise to read, “1.0 mL of fermentation broth, incubated for different incubation times, …”

4.      Legends on Figure 1, 2, and 3 should explain the meaning of the error bars, e.g., “Error bars indicate the standard deviation from the mean of three replicates.”

5.      Figure 2b: was isoamyl alcohol not detected in the “No amino acid” condition? This should be explained in the text or figure caption.

6.      In Figure 2d, the highest titers of octanoate were detected early in fermentation for Phe but later in fermentation for Leu. Why might this be?

7.      Figure 2i,j: were phenylethyl hexanoate and phenylethyl carpylate not detected in the “No amino acid” and “Leu” conditions? This should be explained in the text or figure caption.

8.      Lines 288-290: “It is speculated that the pathway of esters synthesis by acetyl-CoA is not the main biosynthesis pathway for phenylethyl acetate and isoamyl acetate.” Is there evidence to suggest this, aside from the correlation analysis presented here? If so, this statement should include reference(s).

Author Response

Thanks for your suggestion. We have provide a point-by-point response to your comments, and please see the attachment.

Reviewer 2 Report

The work is very interesting, as a general observation, it is important that the authors reinforce their discussion through a greater number of bibliographical references.

Line 93. Was the fermentation without agitation?

Line 117. “One mL fermentation…”

Lines 146-152. Indicate the value of alpha. Did you apply any experimental design? Were the analyzes done in triplicate?

Line 161. Remove the zero

Lines 169-170. Is there any reported metabolic pathway that reinforces this statement?

Figure 2. Why does the control without amino acids not appear in figures 2b, 2i and 2j?

Line 269. Correct “Simulated”

Lines 257-273. Are the observed behaviors comparable with those obtained by other authors?

A graph that replaces tables 1 and 2 of the correlation analysis would better explain the data.

Author Response

(The authors gave the same response as above.)

Round 2

Reviewer 1 Report

I am satisfied with the improvements made to this manuscript, and it is sufficient for publication in Molecules.